# Megalithic statue (*moai*) production on Rapa Nui (Easter Island, Chile)

**Carl Philipp Lipo**[1☯*], **Terry L. Hunt**[2☯], **Gina Pakarati**[3‡], **Thomas Pingel**[4‡], **Noah Simmons**[2‡], **Kevin Heard**[4‡], **Laryssa Shipley**[2‡], **Caroline Keller**[2‡], **Colin Omilanowski**[2‡]

1 Department of Anthropology, Binghamton University, Binghamton, New York, United States of America, 2 School of Anthropology, University of Arizona, Tucson, Arizona, United States of America, 3 Hanga Roa, Rapa Nui, Chile, 4 Department of Geography, Binghamton University, Binghamton, New York, United States of America

☯ These authors contributed equally to this work.
‡ GP, TP, NS, KH, LS, CK and CO also contributed equally to this work.
* clipo@binghamton.edu

## Abstract

Ethnohistoric and recent archaeological evidence suggest that Rapa Nui (Easter Island, Chile) was a politically decentralized society organized into small, relatively autonomous kin-based communities across the island. The more than 1,000 monumental statues (*moai*) of Rapa Nui thus raise a critical question: was production at Rano Raraku—the primary *moai* quarry—centrally controlled or did it mirror the decentralized pattern found elsewhere on the island? Using Structure-from-Motion (SfM) photogrammetry with over 11,000 UAV images, we created the first comprehensive three-dimensional model of the quarry to test these competing hypotheses. Our analysis reveals 30 distinct quarrying foci distributed across the crater, each containing redundant production features and employing varied carving techniques. This spatial organization, combined with evidence for multiple simultaneous workshops constrained by natural boundaries, indicates that *moai* production followed the same decentralized, clan-based pattern documented for other aspects of Rapa Nui society. These findings challenge assumptions that monumentality requires hierarchical control, instead supporting emerging frameworks that recognize how complex cooperative behaviors can emerge through horizontal social networks. The high-resolution 3D model also establishes a crucial baseline for the cultural heritage management of this UNESCO World Heritage site, while advancing methodological approaches for testing sociopolitical hypotheses through the spatial analysis of archaeological landscapes.

**Data availability statement:** All feature data and the 3D model generated and analyzed in this study are publicly available Zenodo at https://doi.org/10.5281/zenodo.17095895. An interactive version of the 3D model is available at. https://arcg.is/qu59O1.

**Funding:** Funding for the fieldwork was supported by a National Science Foundation grant (Award #2218602). The funders had no role in study design, data collection and analysis, decision to publish, or preparation of the manuscript.

**Competing interests:** The authors have declared that no competing interests exist.

## Introduction

The monumental stone statues (*moai*) of Rapa Nui represent one of Polynesia's most striking archaeological phenomena, with over 1,000 megalithic figures distributed across the small volcanic island of Rapa Nui, which measures only 163.6 square kilometers. This remarkable investment in monumentality appears paradoxical when considered alongside ethnohistoric accounts that consistently describe Rapa Nui society as organized into relatively small, competing kin-based groups rather than a unified polity. Early ethnographers, including Routledge [1], Métraux [2], and Englert [3], documented a sociopolitical landscape characterized by multiple *mata* (clans or tribes) that maintained distinct territorial divisions, separate ceremonial centers, and autonomous leadership structures. These accounts suggest a society lacking centralized authority, despite having produced one of Polynesia's most extensive programs of monumentality.

Recent archaeological investigations have provided a growing body of evidence in support of decentralized social organization. Analyses of settlement patterns reveal spatially discrete community clusters with redundant architectural features, suggesting autonomous residential groups rather than hierarchical integration [4]. Studies of obsidian tool distributions reveal highly localized patterns of material culture, indicating limited inter-community interaction [5]. Transport road analyses reveal radial patterns emanating from the quarry, suggesting independent group efforts rather than coordinated infrastructure [6,7]. Experiments and analyses of *moai* transport by 'walking' reveal that relatively small work groups of 15–50 would have been sufficient to move even the largest *moai* [8]. Studies of *pukao* (red scoria cylinder 'hats') placed atop many *moai* similarly required teams of only about 15 people to position even the largest ones [9,10]. Spatial associations between monuments and resources follow territorial boundaries consistent with clan-based control [11], while population modeling indicates that cultural patterns emerged from interactions among small, semi-isolated communities [12]. This convergence of evidence points toward a society organized around multiple small-scale groups that maintained cultural cohesion while operating with considerable autonomy.

One critical question remains unresolved in this emerging picture of Rapa Nui social organization: does the pattern of decentralized, clan-based structure extend to the island's primary *moai* production center at Rano Raraku? The quarry presents a unique context for examining sociopolitical organization, as it represents both a singular resource, the island's only source of suitable volcanic tuff for statue carving, and the locus of the most labor-intensive component of monument construction. If *moai* production was centrally controlled, we expect to find evidence of standardized production methods, hierarchical spatial organization, and restricted access, indicating top-down management of this critical resource. Alternatively, if the decentralized pattern observed elsewhere on the island extended to the quarry, we would expect to find multiple groups working simultaneously in discrete areas, employing varied techniques, and maintaining the competitive yet culturally unified dynamics known ethnographically. For this study, we define "centralization" as a system of authority

consolidated in a single political unit that exerts exclusive control over access to critical resources and labor organization at the island-wide or supra-local community scale. By contrast, "decentralization" refers to multiple autonomous groups retaining control over production without evidence of a dominant authority imposing its will on production. We recognize the possibility that kin groups could centralize their own efforts at localized scales, but our analysis evaluates whether the quarry reflects integration under a singular coordinating authority or, instead, a mosaic of parallel, semi-autonomous operations. Such questions lie at the heart of understanding how societies might coordinate monumental investments without hierarchical political structures—a pattern with implications extending far beyond Rapa Nui.

## Rapa Nui and the Rano Raraku quarry

Rapa Nui is a small (163.6 km$^2$) volcanic island located in the southeastern Pacific, approximately 3,500 km west of the coast of South America and 4,260 km southeast of Tahiti (Fig 1). The island consists of a single landmass formed from three volcanoes: Poike, Rano Kau, and Terevaka. The archaeological record of Rapa Nui reveals a Polynesian settlement that dates to the 13th century CE [13–17]. The island's archaeological record reveals dispersed community patterning centered on the coasts, widespread cultivation, and, most famously, monumental statuary and architecture. *Moai* are monolithic statues carved from stone that were transported across the island and erected on ceremonial platforms (*ahu*)

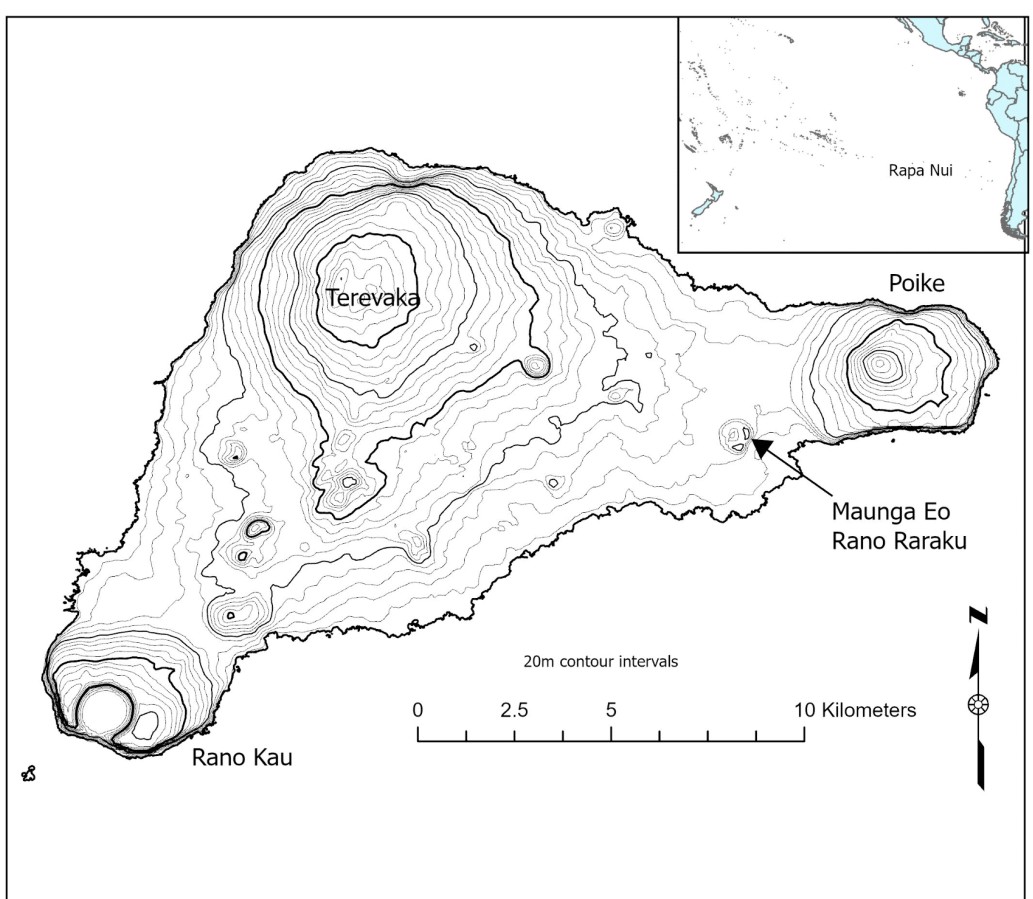

**Fig 1. Geographic context of Rapa Nui.** Map of Rapa Nui with Maunga Eo and Rano Raraku quarry indicated. (Inset) Location of Rapa Nui in the eastern Pacific Ocean highlighting its extreme isolation.

by Rapa Nui communities between the 13th and 17th centuries CE. Rano Raraku stands as the primary quarry site for *moai* production. The Rano Raraku *moai* quarry is situated on the volcanic crater, traditionally known as Maunga Eo, a pyroclastic cone formed from a vent of the Poike volcano during its caldera stage, approximately 0.4 to 0.3 million years ago [18]. This eruption produced hyalotuff, fine volcanic glass fragments that form when magma explosively erupts in the shallow ocean [19,20]. Once the pyroclastic cone emerged from the ocean, it was ultimately exposed to the sea, with erosion creating a cliff on the southeast-facing side. Later, lava flows from Terevaka surrounded Maunga Eo as the island's three volcanoes merged to form the island.

The Maunga Eo volcanic cone is the source of over 95% of the island's more than 1,000 iconic statues. Polynesian settlers began carving these massive figures shortly after the island's colonization in the 13th century CE [14–16,21,22]. They transported *moai* across the rocky terrain to stand on monumental constructed stone platforms known as *ahu* [23]. The Rano Raraku quarry has hundreds of *moai* found in various stages of completion, with the most iconic standing partially buried along the lower slopes of the volcanic cone.

The documentation of the Rano Raraku quarry has a long history, beginning with Lieutenant-Captain Geiseler's 1882 descriptions [24] and Thomson's [25] first systematic archaeological investigations, which recorded 155 images on the western exterior slopes. Katherine Routledge [1] conducted the first comprehensive survey and mapping in 1914, inventorying 293 *moai* [26]. Subsequent efforts by Métraux [2], Cornejo and Atan [26], Englert [3], and Skjølsvold [27] yielded varying *moai* counts, ranging from 157 to 297. While Cristino et al. [28] created the first systematically generated and detailed map of the quarry as part of their *Atlas arqueológico de Isla de Pascua*, complete documentation from this ambitious work is not publicly available, limiting its utility in further research. Van Tilburg's documentation project, begun in 1981−82 and continuing through at least 2019, inventoried 406 sculpted objects and included details about the quarry. While her dissertation data provides measurements for 245 *moai,* many of which are located in the quarry [29]*,* the results of the quarry investigations are currently unavailable for open research. Researchers such as Hamilton and Richards have explored the symbolic and ritual aspects of past quarrying activities, primarily focusing on the *moai*-related activities surrounding the quarry [30–32]. Efforts by Riquelme [33], Shepardson [34,35], and Hochstetter and colleagues [36] aimed to create publicly available inventories and descriptions of *moai.* The combined dataset of these studies, produced by Schumacher [37], however, provides only limited descriptions of statues in the quarry.

Despite extensive documentation efforts spanning over 140 years, systematic, accessible data for Rano Raraku remain piecemeal and inconsistent, with varying counts, incomplete publications, and inaccessible databases hampering comprehensive research and cultural heritage management. While various researchers have collected partial measurements and locations for some *moai*, from Routledge's [1] early mapping to Van Tilburg et al.'s GPS positioning and Schumacher's [37] combined inventory, we still lack a systematic, comprehensive map of the quarry and complete three-dimensional documentation of the site. More critically, we know little about the contextual relationships of *moai* within the quarry: their orientation relative to geological bedding planes, the spatial organization of carving activities, the sequential patterns of production, or the technological decisions that guided where and how statues were carved from the tuff. Although researchers like Skjølsvold [27] and Métraux [2] provided observations about carving procedures and the variable orientations of statues relative to the quarry slope, these remain anecdotal descriptions rather than systematic documentation. The absence of comprehensive spatial and contextual data greatly limits our ability to test hypotheses about quarrying strategies, production sequences, or aspects of the social organization of carving activities. In addition, without detailed three-dimensional documentation that captures both the morphology of individual *moai* and their relationships to the quarry landscape, we cannot deploy modern analytical approaches, such as morphometric analyses, spatial statistics, or computational modeling, that could reveal new insights into the evolution of statue forms, standardization in production, or the decision-making processes that shaped this remarkable archaeological landscape.

Despite its significance, documenting the quarry is challenging due to its complex three-dimensional landscape and vast scale. To address this challenge, we employ UAV-based Structure-from-Motion (SfM) photogrammetry, utilizing over

10,000 images captured by the UAV, to generate the first comprehensive high-resolution three-dimensional model of the quarry at Rano Raraku [38]. This innovative approach overcomes the limitations of traditional documentation methods, enabling us to quantify production features and identify spatial patterns that suggest the social organization behind *moai* production. Our three-dimensional model suggests that statue production was organized into multiple, likely simultaneously active workshop areas, raising questions about the scale of production that aligns with evidence for a decentralized, clan-based sociopolitical structure on Rapa Nui.

## Materials and methods

Between June 2023 and January 2024, we conducted a series of low-elevation (30 and 50m) flights using DJI Mavic 3 Enterprise UAVs equipped with 20-megapixel CMOS sensors and 24 mm lenses. Our work was requested and authorized by the Rapa Nui *Comunidad Indígena Ma'u Henua*. All necessary permits were obtained for the described study, which complied with all relevant regulations. We flew the UAVs at 30m AGL using active terrain-following, which enabled consistent ground resolution despite Rano Raraku's steep slopes and varying elevations (ranging from 70 to 150m above sea level). We captured 11,686 photographs with 80% overlap and sidelap, combining nadir and 45-degree oblique orientations with double grid patterns to ensure robust photogrammetric reconstruction. Our flight paths over the quarry are documented in S1 Fig. The resulting images have a pixel resolution of 0.86–1.12 cm, which is sufficient to document subtle archaeological features.

We processed these images using ESRI's ArcGIS Reality for ArcGIS Pro [39], which employs Structure-from-Motion (SfM) algorithms to assemble three-dimensional structures through the Scale-Invariant Feature Transform (SIFT) [40] This software's ability to handle datasets exceeding 300 gigapixels while maintaining geometric accuracy was crucial for our extensive coverage.

The resulting Reality Mesh format efficiently places more faces where intricate detail is required and fewer faces in flat areas, allowing for highly detailed and flexible three-dimensional models (Fig 2). The full, interactive model is publicly accessible via ArcGIS Online ESRI's cloud-based mapping platform: https://arcg.is/qu59O1.

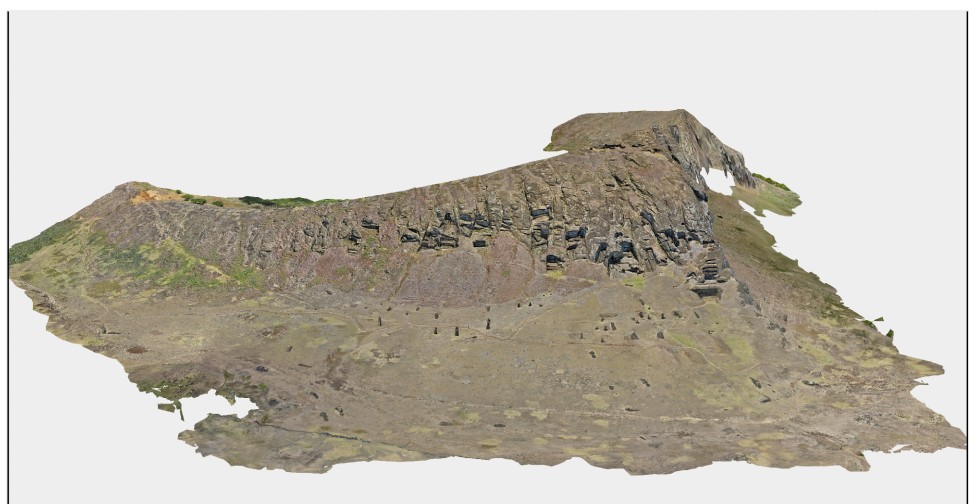

**Fig 2. Three-dimensional model of Rano Raraku quarry produced through Structure-from-Motion photogrammetry.** This comprehensive digital documentation, derived from 11,686 UAV images, reveals the complex spatial organization of production activities distributed across multiple workshop areas.

## Results

The high-resolution, three-dimensional model provides unprecedented documentation of *moai* and quarrying features (Figs 3–5), as well as previously unrecorded features and spatial patterns difficult to discern through traditional mapping techniques.

### Comprehensive inventory of production features

From our analysis, we have recorded 426 features representing *moai* in various stages of completion, 341 trenches cut to outline blocks for carving, 133 quarried voids where statues were successfully removed, and five bollards that likely served as anchor points for lowering *moai* down slopes (Table 1). The model also revealed features, including quarrying areas on the crater's exterior slope (Fig 6) and a system of bollards associated with *pu* (large bedrock pits) that likely facilitated the transport of *moai* down steep terrain (Figs 4 and 7). Additional features will become visible upon further inspection of the model and ongoing photographic monitoring.

### Multiple carving methods

The model demonstrates that at least three distinct quarrying procedures were employed across the volcanic crater landscape (Fig 8, Table 2). The most common method (143 examples) involved defining facial details before outlining the head and body. In 120 cases, blocks were completely outlined before detailed carving began. In five instances where cliff faces were near vertical, carvers worked sideways into the cliff. The evidence in the quarry shows that initial carving most often began with trenches cut into the bedrock, which, as the process continued, created rectangular blocks. We could think of

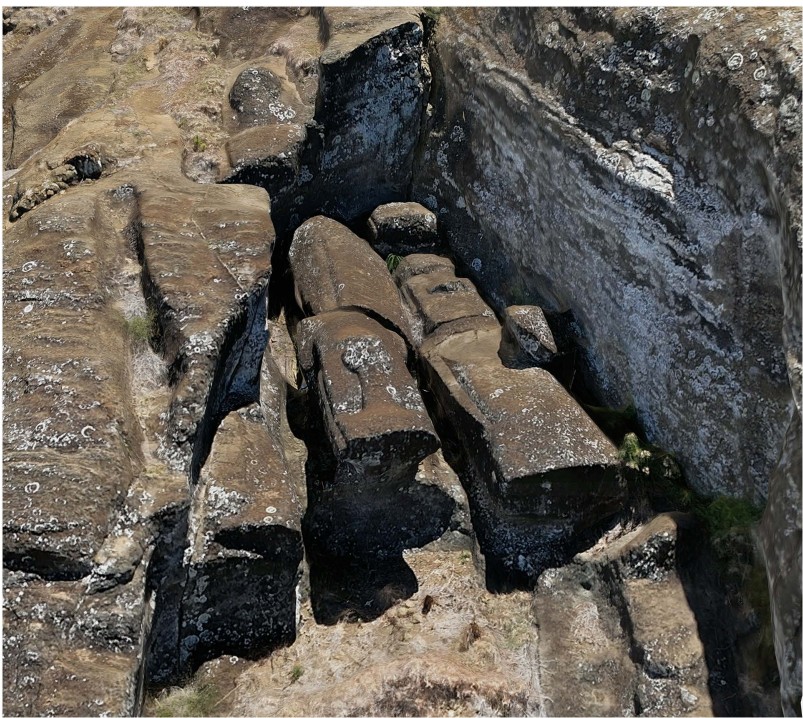

**Fig 3. Production technique revealed through 3D modeling.** Unfinished *moai* attached to bedrock by "keels" along their backs demonstrate how carvers worked underneath from both sides until figures were separated from the source material. This production stage, difficult to document through traditional methods, is visible in the 3D model.

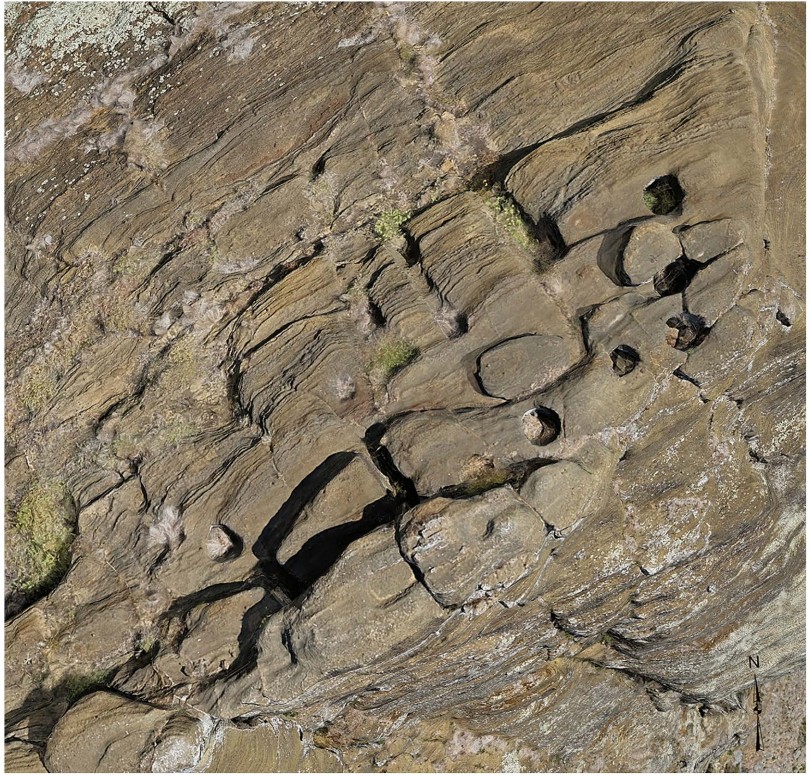

**Fig 4. Engineering infrastructure supporting decentralized production.** *Pu makari* features (carved holes) and anchor stones near the summit of *Maunga Eo* facilitated the movement of statues by small work groups, eliminating the need for centralized coordination of large labor forces.

these as "*moai* preforms." Depending on the specific location within the quarry, statues were carved in horizontal, vertical, or inclined orientations, with some positioned in stepped arrangements along the quarry slopes, as observed historically [1–3]. Most *moai* were quarried and carved oriented in supine position, with the majority carved from the top down, while a smaller number were extracted and shaped from the side. This diversity of approaches suggests adaptation to local geological conditions rather than a standardized, centrally managed production system. While variation in carving techniques does not preclude centralized organization, stylistic or procedural diversity can emerge through workshop traditions, family units, or individual artisanship. What distinguishes the Rano Raraku case is the spatial redundancy of multiple workshop foci operating contemporaneously without evidence of overarching standardization, restricted access, or hierarchical oversight. We suggest this distributed pattern is more consistent with coordination at the level of clans than with a single dominant polity.

## Scale of *Moai* production

Significantly, our model reveals at least 30 distinct quarrying foci distributed across the crater landscape (Figs 9 and 10). These workshops are identified as spatially continuous areas of redundant quarrying activity, typically delineated by unquarried bedrock or natural discontinuities. Each focus contains a combination of voids where statues were removed, and figures abandoned in various stages of completion.

Physical constraints within each workshop area limited the number of carvers who could work simultaneously, suggesting relatively small production teams [1], with perhaps 4–6 active carvers engaged directly on a single *moai* at one time,

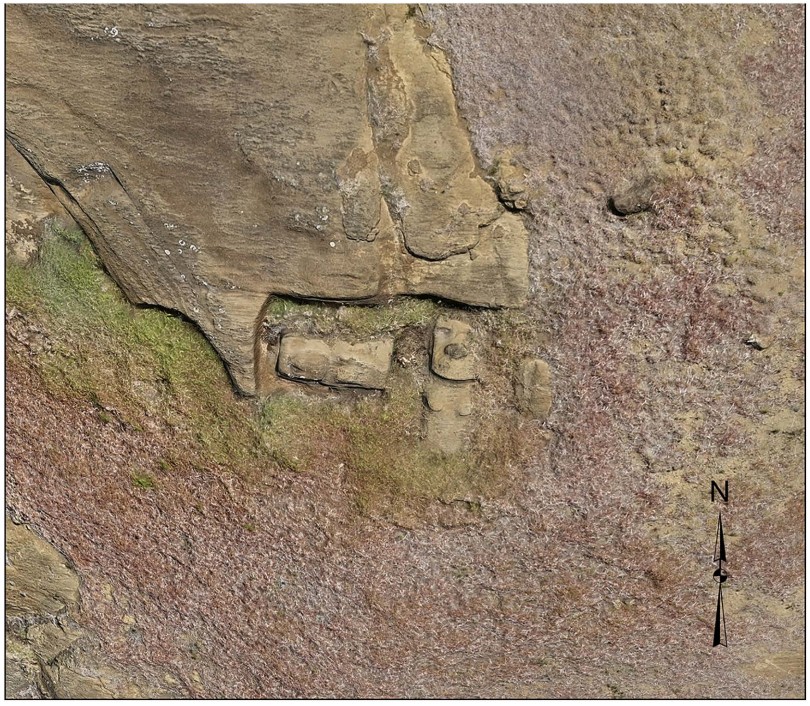

**Fig 5. Artistic variation within standardized forms.** Female *moai* on the upper slopes of Rano Raraku demonstrate how distinct workshop areas maintained some stylistic autonomy while adhering to general design conventions, consistent with distributed production by semi-autonomous groups.

**Table 1. Features identified at Rano Raraku.** A comprehensive inventory of archaeological features revealed through 3D modeling, showing the diversity and quantity of production elements across 30 distinct quarrying foci. This quantification demonstrates the distributed nature of *moai* production.

| Feature | Count | Notes |
| --- | --- | --- |
| Sculpted objects | 426 | Areas of carving that are part of shaping a *moai*. These areas include blocks shaped to carve *moai* "pre-forms," *moai* still connected to the bedrock, and figures left unfinished or in the process of transport. |
| Trenches | 341 | Linear excavation to test volcanic tuff and/or rough out *moai* "preforms." |
| Quarried voids | 133 | Roughly rectangular voids in the bedrock resulting from the removal of *moai*. |
| Quarrying foci | 30 | Distinct clusters of quarrying activity that reflect the foci of carver groups. |
| *Pu* | 13 | Round holes in which large posts may have been placed to serve as anchor points for ropes. |
| *Taheta* | 7 | Basins carved into the bedrock for rainwater catchment. |
| Bollards | 5 | Carved areas of bedrock that may have served as anchors to control ropes used to lower carved tuff blocks. |
| Other features | 17 | Includes 11 areas of quarried waste material, 3 *umu* (earth ovens), 1 *hare paenga* (house foundation), 1 built earth and stone ramp, and 1 rock alignment. |

with additional personnel (perhaps 10–20 more) supporting activities such as tool and rope production, and provisioning. Thus, while the carving locus itself restricted active workers, broader production crews likely operated at a somewhat larger scale, consistent with kin-based groups rather than an island-wide workforce. The redundancy of these work areas, combined with the relative stylistic uniformity of the statues, points to contemporaneous rather than sequential operation by multiple independent groups. Such a scale of *moai* production by clan-based groups (or "tribes," *mata*) conforms to patterns of social organization described by earlier researchers [1–3].

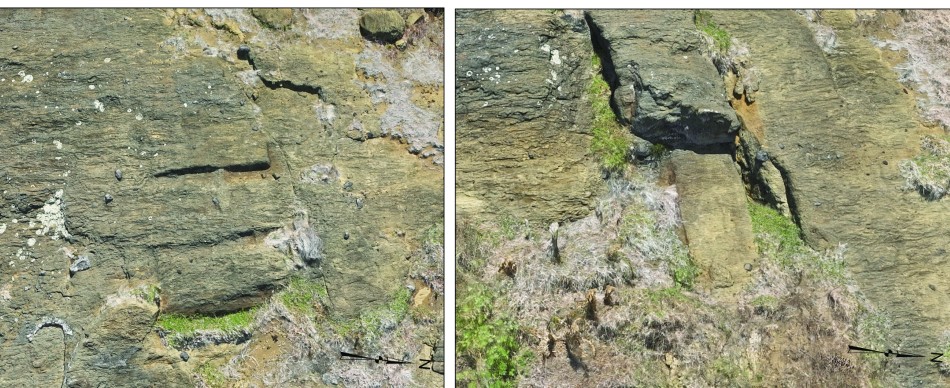

**Fig 6. Previously undocumented production area.** Newly identified trenches on the southeast ridgeline, revealed through comprehensive 3D mapping, extend the known distribution of quarrying activity.

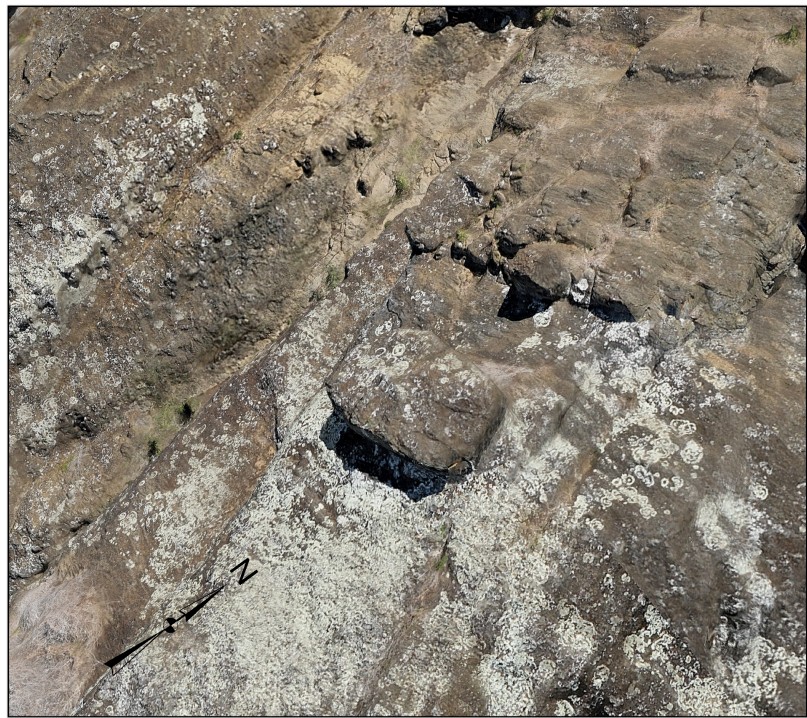

**Fig 7. Bollard feature, southeast slope.**

## Discussion

Our findings suggest a decentralized network of production, consistent with a sociopolitical organization comprising multiple small, interacting kin-based communities rather than a more unified authority structure, aligning with broader ethnographic and archaeological evidence from the island. As we have outlined, the *moai* transport routes show that paths radiated outward from Rano Raraku in patterns suggesting separate group efforts rather than a coordinated system [6]. Investigations of statue transport mechanisms demonstrate that relatively small groups could efficiently move even the

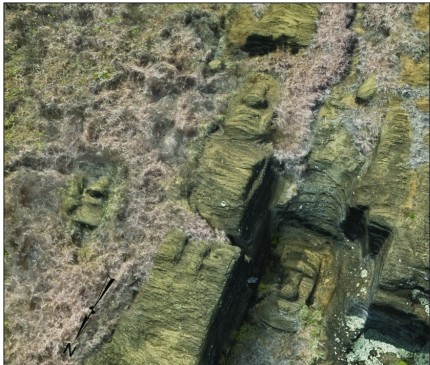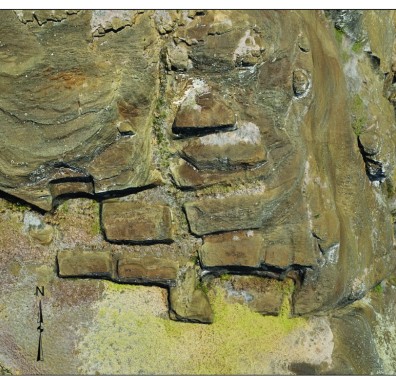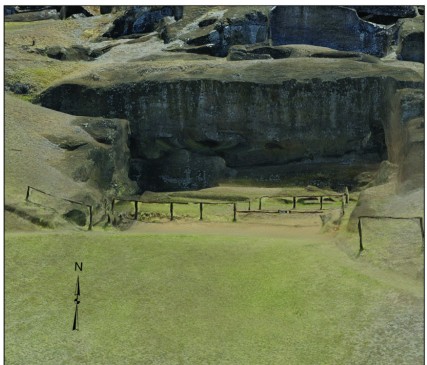

**Fig 8. Diverse production methods were identified across workshop areas.** Three distinct carving approaches: (Left) Face-first method, where facial features are outlined before the body; (Middle) Block method, where complete outlines are created on all sides; (Right) Sideways method for vertical cliff faces. This methodological diversity suggests adaptation by autonomous production groups.

**Table 2. Quarrying methods employed at Rano Raraku. Three distinct carving approaches were identified across the quarry, indicating that autonomous production groups adapted to local conditions rather than standardized procedures imposed by centralized management.**

| Method | Count | Notes |
|---|---|---|
| Face Outline | 143 | *Moai* are carved in a supine position. The face and head outlines are carved in detail, followed by the body. |
| Block | 120 | Blocks of volcanic tuff are outlined first; then, body outlines are carved, followed by faces. |
| Lateral | 5 | Carving is conducted into the vertical faces of the cliff. The upper part of the body is outlined, and the details of the face are carved first. This number includes the initial carving of an experimental *moai* attempted by Heyerdahl (39). |

largest statues without centralized coordination. Morrison's [4] analysis of community architecture reveals spatially clustered, functionally redundant complexes of limited scale, consistent with organization around small, relatively autonomous kin groups [4–7,12]. Rapa Nui is not unique in this regard, as elsewhere in Polynesia we find aggregative productive activities organized through distributed networks of semi-autonomous groups, without the direction of a paramount authority. For example, the early phases of large-scale agricultural field systems of Kona and Kohala, Hawai`i Island, likely emerged in the absence of centralized control [41,42].

While communities maintained independent artistic expression in architectural forms and material culture across the island, the statues carved at Rano Raraku during peak production converged on standardized forms. This standardization within clan-based production suggests information sharing and cultural connectivity between groups in the absence of centralized hierarchical control. Our findings thus support archaeological and biological evidence indicating high local cultural retention with limited island-wide community interaction [12].

The organizational pattern revealed in multiple lines of evidence, including our findings reported here, has significant implications beyond Rapa Nui archaeology. As Graeber and Wengrow [43] argue, decentralized political arrangements have been common throughout human history but are often overlooked in favor of assumptions about inevitable hierarchy. Rapa Nui provides a case that challenges the assumption that monumental architecture requires strongly hierarchical political structures. This finding contributes to growing evidence that complex productive activities and large-scale cooperation can emerge through horizontal social networks rather than vertical power structures [7,11,44–46]. Interestingly, a pattern of competitive groups is echoed in the structure of the archaeological record in Orongo, the site of the historic

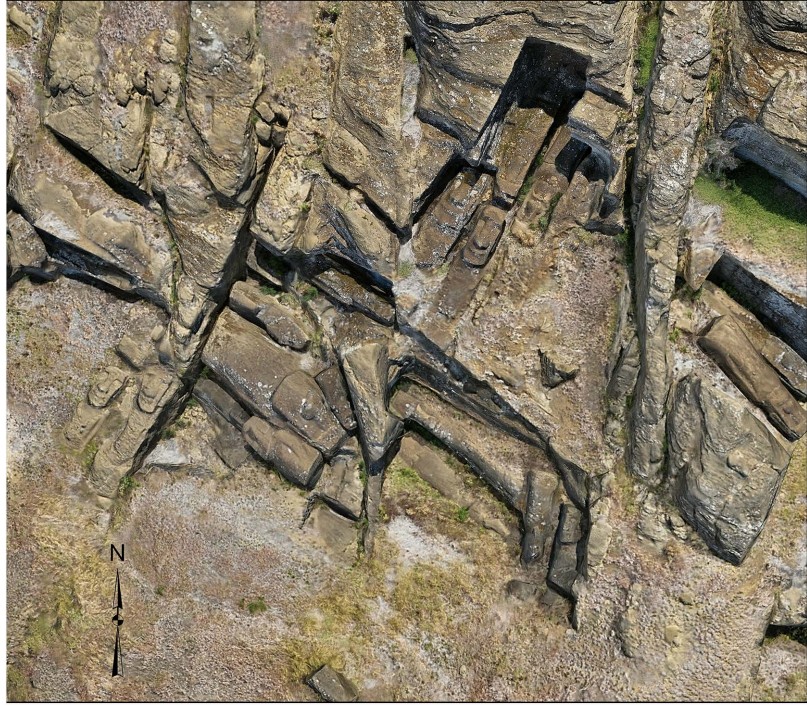

**Fig 9. Boundary identification between workshop areas.** High-resolution 3D modeling enables precise delineation of production zones, revealing patterns of spatial organization that support models of decentralized rather than hierarchical production.

*Tangata Manu* (Birdman) competition in the southwest corner of the island [47]. Moreover, a decentralized production network of *moai* production suggests a sociopolitical system that successfully coordinated monumental construction while avoiding a central concentration of power that could potentially lead to resource overexploitation.

Beyond its archaeological significance, our methodological approach offers advantages for the management of cultural heritage. The comprehensive three-dimensional model is a baseline for monitoring erosion, climate impacts, tourism effects, and site degradation. Following a 2022 grass fire that threatened the quarry, local heritage managers (*Comunidad Indígena Ma'u Henua*) requested this documentation to assess damage and develop protection strategies. Unlike previous research that often produced limited datasets, our model is publicly accessible through ArcGIS Online, enabling stakeholders to monitor changes and plan conservation with greater precision.

## Conclusions

The identification of 30 distinct quarrying foci, each containing redundant production features and employing varied carving methods, demonstrates that multiple groups worked simultaneously at the quarry without apparent centralized control. This pattern directly parallels the territorial divisions, autonomous community clusters, and competitive dynamics described by Routledge [1], Métraux [2], and Englert [3], extending the evidence for decentralized organization to the island's most significant production center. The spatial organization of these workshop areas, physically constrained by natural boundaries and unquarried bedrock, suggests that a controlling authority did not restrict access to the quarry but instead negotiated among multiple participating groups. The diversity of carving techniques, from face-first to block methods, indicates technological autonomy rather than standardized procedures that would be expected under hierarchical management. Yet, despite this organizational independence, the stylistic convergence of *moai* forms demonstrates

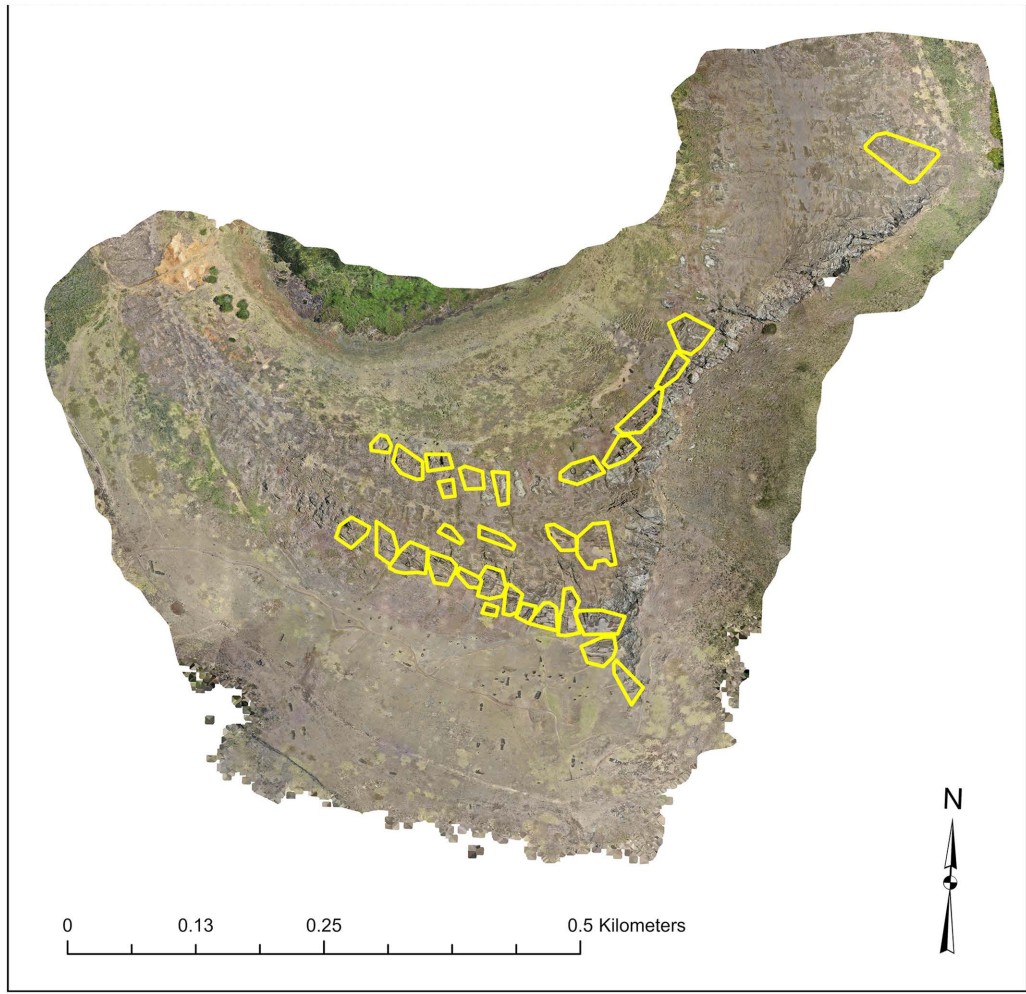

**Fig 10. Network of 30 distinct quarrying foci across Rano Raraku.** Yellow polygons outline the distribution of separate workshop areas, each containing redundant sets of production features. This pattern supports hypotheses of the decentralized sociopolitical organization of *moai* production.

the powerful role of cultural tradition and information sharing in coordinating production without centralized oversight. This finding reinforces emerging theoretical frameworks that recognize how monumental construction can emerge through horizontal social networks rather than vertical power structures [7,11,12,43,44,46].

The implications extend beyond Rapa Nui to challenge assumptions about the relationship between monumentality and political hierarchy. Our results demonstrate that societies can successfully coordinate large-scale productive activities through distributed networks of semi-autonomous groups, maintaining both competitive dynamics and cultural cohesion. This decentralized production system may have been advantageous, preventing the concentration of power that could lead to resource overexploitation while enabling flexible responses to environmental and social challenges. The Rano Raraku evidence thus contributes to a growing recognition that complex cooperative behaviors and monumental achievements can emerge from heterarchical rather than hierarchical social arrangements.

The advantage of documenting the archaeological record in three-dimensional models is evident in the case of the Rano Raraku quarry. As *moai* quarrying took place across and into the face of the sloping cliff at various angles, two-dimensional renderings, such as plans or profiles, cannot capture the details of these archaeological phenomena. The

quarry's complex topography, with its multiple working faces, varying slopes, and intricately carved features, creates a three-dimensional puzzle that defies traditional documentation methods. The overlapping nature of quarrying activities, where later extractions often intersect with earlier ones, further complicates efforts that rely on documentation in two dimensions.

Previous solutions, such as the renderings rendered by Van Tilburg and colleagues [48], offer one way of addressing this problem through detailed artistic interpretations. Drawings, however, are inherently interpretive, dependent on the viewer's perspective and understanding. In contrast, the models provide measurable, geometrically accurate representations that can be examined from any angle, under varying lighting conditions, and at different scales. With these photo-realistic three-dimensional models, researchers can explore details from multiple angles and perspectives, enabling new forms of analysis and verification. This flexibility is valuable for studying the quarry's complex production sequence and spatial organization.

The digital nature of these models enables sophisticated analytical techniques that were previously impossible with traditional documentation methods. Researchers can now analyze shape variability, generate precise measurements, extract cross-sections, and conduct volumetric calculations, enabling quantitative analysis of quarrying techniques and production strategies. This study demonstrates that computational approaches to archaeological documentation can fundamentally reshape our understanding of ancient production systems and their socio-political implications. With these new, publicly available data, researchers can begin to evaluate hypotheses about aspects of Rapa Nui's past that were previously untestable. The ability to model complex three-dimensional relationships between archaeological features, for example, enables the identification of spatial patterns that reveal organizational principles invisible through traditional methods. This approach has applications well beyond Rapa Nui, offering new methodological possibilities for archaeology worldwide.

The ability to test long-standing questions about Rapa Nui's social organization through comprehensive spatial documentation underscores the transformative potential of new archaeological methods. Where previous researchers could only speculate about quarry organization based on limited observations, we can now quantify the distribution of workshop areas, analyze production strategies, and identify organizational patterns that directly address hypotheses about sociopolitical structure. This study demonstrates that fundamental questions about past societies—questions that have implications for understanding human cooperation, monument building, and political organization more broadly—can be revisited and resolved through innovative approaches to archaeological documentation.

## Supporting information

**S1 Fig. Methodological approach for comprehensive site documentation.** UAV flight paths demonstrating systematic coverage at 30 meters above ground with terrain-following capability. Nadir and oblique photos with 80% overlap enable accurate 3D reconstruction of complex archaeological features, transforming capabilities for documenting and analyzing ancient production landscapes.
(TIF)

## Acknowledgments

The authors thank the members of the 2023 and 2024 *Comunidad Indígena Ma'u Henua* Board of Directors. ESRI Customer Support provided invaluable assistance with the 3D modeling.

## Author contributions

**Conceptualization:** Carl Philipp Lipo, Terry L. Hunt, Gina Pakarati, Kevin Heard.

**Data curation:** Carl Philipp Lipo.

**Formal analysis:** Carl Philipp Lipo, Noah Simmons, Caroline Keller, Colin Omilanowski.

**Funding acquisition:** Carl Philipp Lipo.

**Investigation:** Carl Philipp Lipo, Terry L. Hunt, Gina Pakarati, Noah Simmons, Laryssa Shipley.

**Methodology:** Carl Philipp Lipo, Terry L. Hunt, Gina Pakarati.

**Project administration:** Carl Philipp Lipo, Terry L. Hunt, Gina Pakarati.

**Resources:** Carl Philipp Lipo, Terry L. Hunt, Gina Pakarati.

**Software:** Carl Philipp Lipo, Thomas Pingel, Noah Simmons, Kevin Heard.

**Supervision:** Carl Philipp Lipo, Terry L. Hunt.

**Validation:** Carl Philipp Lipo, Gina Pakarati, Thomas Pingel.

**Visualization:** Carl Philipp Lipo, Thomas Pingel, Noah Simmons, Kevin Heard, Laryssa Shipley, Caroline Keller, Colin Omilanowski.

**Writing – original draft:** Carl Philipp Lipo, Terry L. Hunt.

**Writing – review & editing:** Carl Philipp Lipo, Terry L. Hunt, Thomas Pingel, Noah Simmons, Kevin Heard.

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
