## [Decision Letter · Decision Letter 0]

1 Sep 2025

Dear Dr. Lipo,

Thank you for submitting your manuscript to PLOS ONE. After careful consideration, we feel that it has merit but does not fully meet PLOS ONE’s publication criteria as it currently stands. Therefore, we invite you to submit a revised version of the manuscript that addresses the points raised during the review process.

Please address all comments by the reviewer in detail before re-submission

We look forward to receiving your revised manuscript.

Kind regards,

Peter F. Biehl, PhD

Academic Editor

PLOS ONE

Journal Requirements: 

2. In your manuscript, please provide additional information regarding the specimens used in your study. Ensure that you have reported human remain specimen numbers and complete repository information, including museum name and geographic location.

For more information on PLOS One's requirements for paleontology and archeology research, see https://journals.plos.org/plosone/s/submission-guidelines#loc-paleontology-and-archaeology-research .

3. Please include a complete copy of PLOS’ questionnaire on inclusivity in global research in your revised manuscript. Our policy for research in this area aims to improve transparency in the reporting of research performed outside of researchers’ own country or community. The policy applies to researchers who have travelled to a different country to conduct research, research with Indigenous populations or their lands, and research on cultural artefacts. The questionnaire can also be requested at the journal’s discretion for any other submissions, even if these conditions are not met.  Please find more information on the policy and a link to download a blank copy of the questionnaire here: https://journals.plos.org/plosone/s/best-practices-in-research-reporting. Please upload a completed version of your questionnaire as Supporting Information when you resubmit your manuscript.

 [Funding for the fieldwork was supported by a National Science Foundation grant (Award #2218602).]. 

5. Thank you for uploading your study's underlying data set. Unfortunately, the repository you have noted in your Data Availability statement does not qualify as an acceptable data repository according to PLOS's standards.

6.  We note that Figure(s) 1, 2, 9 and S1  in your submission contain [map/satellite] images which may be copyrighted. All PLOS content is published under the Creative Commons Attribution License (CC BY 4.0), which means that the manuscript, images, and Supporting Information files will be freely available online, and any third party is permitted to access, download, copy, distribute, and use these materials in any way, even commercially, with proper attribution. For these reasons, we cannot publish previously copyrighted maps or satellite images created using proprietary data, such as Google software (Google Maps, Street View, and Earth). For more information, see our copyright guidelines: http://journals.plos.org/plosone/s/licenses-and-copyright.

1. You may seek permission from the original copyright holder of Figure(s) 1, 2, 9 and S1  to publish the content specifically under the CC BY 4.0 license. 

7. We notice that your supplementary figure is included in the manuscript file. Please remove them and upload them with the file type 'Supporting Information'. Please ensure that each Supporting Information file has a legend listed in the manuscript after the references list.

Additional Editor Comments:

Please address all comments by the reviewer in detail before resubmission

Reviewers' comments:

Reviewer's Responses to Questions

**Comments to the Author**

1. Is the manuscript technically sound, and do the data support the conclusions?

Reviewer #1: Yes

2. Has the statistical analysis been performed appropriately and rigorously?

Reviewer #1: Yes

3. Have the authors made all data underlying the findings in their manuscript fully available?

Reviewer #1: Yes

4. Is the manuscript presented in an intelligible fashion and written in standard English?

Reviewer #1: Yes

Reviewer #1: I thought this was a really interesting paper that accomplishes the major goal of creating a publicly accessible 3D database of the Rano Raraku quarry site on Rapa Nui—and through the collection of high-resolution imagery—postulates how the famous moai statues were carved and the sociopolitical mechanisms behind their construction. Overall, I thought the paper was well written and have only a few, mostly minor, comments that I think would help to improve the paper.

Abstract – the authors pose the paradox first without presenting the underlying question behind it. Logically, it would make more sense to flip the first and second sentences and rephrase slightly.

While the authors provide a brief background section on Rapa Nui and the quarry, for readers unfamiliar with anything but what these statues look like, it would be helpful to provide a brief summary on what we know archaeologically about the colonization of Rapa Nui and the major cultural stages. Included in this should be some temporal inferences that provide context on when the island was first settled, when quarrying began (and at what quarries) and ended, where these statues were eventually placed around the island, etc. This doesn’t have to be lengthy (and the paper is relatively short, so there would seem to be plenty of room to add this in), and some kind of cultural chronology would help many readers (re)familiarize themselves with Rapa Nui’s history which the authors seem to take for granted that everyone knows.

In terms of the centralization question and its role in moai manufacture, I had a few comments/queries. The first is, even in a centralized organization, can there not be variation in what is produced? The authors make the argument that what the detailed imagery (and previous inferences from other scholars) shows is not a hierarchical nature but one that is instead heterarchical. That may very well be the case, but there are many past and present analogues that demonstrate variable ways for producing or constructing objects (apart from mass production where there are templates/casts, etc.).

For example, in many societies that have centralized power (whether big or small), the ways in which megaliths, pottery, stone tools, textiles, etc. were produced can have inherent variation in style and decoration that is reflective of family units, clan/kin, or artisanry groups. I guess that I do not necessarily see the impossibility of this in the Rapa Nui case, despite other lines of evidence suggesting that decentralized power was perhaps the norm.

p. 10 – what is meant by “small production teams”? Are we talking 3 people? 10? 25? And while the workshop area might have certainly limited those who had direct access to the statue being carved, I would imagine that others would be in line to take over during rest breaks, etc. Production crews would also presumably involve others who would provide resources (food, water), etc. I guess my point here is, though the physical space to carve might be constrained, this does not necessarily equate to “smaller production teams” which could be X number of times bigger.

p. 11 – back to the question of centralized coordination and “relatively small groups”, what is meant by “small” and cannot centralization be at the level of smaller (micro) or medium (meso) sized groups (however defined)? I guess the issue for me is that the authors are painting a picture of centralization that assumes an island-wide centralized effort, but a strict definition of the term does not require a large island-wide/village sized group as a requirement for centralized power. Hypothetically, this requirement could technically be met by one village of 100 people or less consolidating their power to quarry one part of Rano Raraku and a completely separate village doing the same contemporaneously. Each has centralized their efforts, which the same evidence presented in this paper would seem to show.

To remedy this, I think the authors should provide a more detailed explanation of what they mean by centralization and connect this to an expanded background section on Rapa Nui prehistory which could provide some additional information on where villages were located, how many the authors think were occupied at any one time, and by how many people.

p. 11 – authors propose “high local cultural retention with limited long-range community interaction.” On an island that is only about 23 km across, what is meant here by “long-range”? This of course is a relative scale, but when it comes down to communication, this would seem to be a negligible variable.

Finally, are there any corollaries in the Pacific or elsewhere that the authors could provide which would either provide support for their interpretation of societies that can “effectively coordinate large-scale productive activities through distributed networks of semi-autonomous groups, while maintaining both competitive dynamics and cultural cohesion”. It would be helpful for readers to know whether this is a highly unique feature to Rapa Nui, or something that is visible elsewhere (and in what circumstances/contexts).

**Do you want your identity to be public for this peer review?** For information about this choice, including consent withdrawal, please see our Privacy Policy

Reviewer #1: No

---

## [Author Response · Author response to Decision Letter 1]

23 Sep 2025

1. Please note that your Data Availability Statement is currently missing [the repository name and/or the DOI/accession number of each dataset OR a direct link to access each database]. If your manuscript is accepted for publication, you will be asked to provide these details on a very short timeline. We therefore suggest that you provide this information now, though we will not hold up the peer review process if you are unable.

a. We have updated the Data Availability Statement. The data we have used is available at: https://doi.org/10.5281/zenodo.17095895

2. Please remove your figures/ from within your manuscript file, leaving only the individual TIFF/EPS image files. These will be automatically included in the reviewer’s PDF

a. Done. We have uploaded the updated manuscript without the figures/images.

3. Please note that PLOS ONE is unable to publish previously copyrighted maps or satellite images, or images created using proprietary data. For these reasons, we cannot publish images generated by software which copyrights their output (such as Google Maps, Street View, and Earth). In order to use these images in your submission, we require explicit permission from the copyright owner to publish the figures under the CC BY 4.0 license.

At this time, please kindly clarify the following regarding Figures 1:

a) Where did the authors obtain the basemaps/shapefiles in Figures 1?

i. The shapefiles that form the basis of Figure 1 are from https://naturalearth.com. These are in the public domain and a reference to the author is not required. This page details the “terms of use” : https://www.naturalearthdata.com/about/terms-of-use/ This page states, “No permission is needed to use Natural Earth. Crediting the authors is unnecessary. “

b. Please state whether the map have been previously copyrighted to your knowledge.

i. This is a new figure created for this publication and has not been copyrighted to our knowledge in any other publication.

====== earlier comments ====

1. Please ensure that your manuscript meets PLOS ONE's style requirements, including those for file naming. The PLOS ONE style templates can be found...

Done. The manuscript now conforms to PLOS ONE’s style requirements (including file naming).

2. In your manuscript, please provide additional information regarding the specimens used in your study. Ensure that you have reported human remain specimen numbers and complete repository information, including museum name and geographic location.If permits were required, please ensure that you have provided details for all permits that were obtained, including the full name of the issuing authority, and add the following statement:

Done. This statement has been added.

3. Please include a complete copy of PLOS’ questionnaire on inclusivity in global research in your revised manuscript. Our policy for research in this area aims to improve transparency in the reporting of research performed outside of researchers’ own country or community. The policy applies to researchers who have travelled to a different country to conduct research, research with Indigenous populations or their lands, and research on cultural artefacts. The questionnaire can also be requested at the journal’s discretion for any other submissions, even if these conditions are not met. Please find more information on the policy and a link to download a blank copy of the questionnaire here: https://journals.plos.org/plosone/s/best-practices-in-research-reporting. Please upload a completed version of your questionnaire as Supporting Information when you resubmit your manuscript.

Done. We have completed the questionnaire and will be submitting it.

[Funding for the fieldwork was supported by a National Science Foundation grant (Award #2218602).]. Please state what role the funders took in the study. If the funders had no role, please state: ""The funders had no role in study design, data collection and analysis, decision to publish, or preparation of the manuscript.""

Done. This statement has been added.

5. Thank you for uploading your study's underlying data set. Unfortunately, the repository you have noted in your Data Availability statement does not qualify as an acceptable data repository according to PLOS's standards.

Done. We have posted to Zenodo and have noted this in the manuscript. The URI is: 10.5281/zenodo.17095895

6. We note that Figure(s) 1, 2, 9 and S1 in your submission contain [map/satellite] images which may be copyrighted. All PLOS content is published under the Creative Commons Attribution License (CC BY 4.0), which means that the manuscript, images, and Supporting Information files will be freely available online, and any third party is permitted to access, download, copy, distribute, and use these materials in any way, even commercially, with proper attribution. For these reasons, we cannot publish previously copyrighted maps or satellite images created using proprietary data, such as Google software (Google Maps, Street View, and Earth). For more information, see our copyright guidelines: http://journals.plos.org/plosone/s/licenses-and-copyright.

We have addressed this issue. All of our figures are now using content that we created. We have removed any background imagery that would pose copyright issues.

7. We notice that your supplementary figure is included in the manuscript file. Please remove them and upload them with the file type 'Supporting Information'. Please ensure that each Supporting Information file has a legend listed in the manuscript after the references list.

This supplementary material has been removed and now fits the format as required by the PLOS ONE style guide.

Done

Additional Editor Comments:

Please address all comments by the reviewer in detail before resubmission

Reviewers' comments:

Reviewer's Responses to Questions

Comments to the Author

1. Is the manuscript technically sound, and do the data support the conclusions?

Reviewer #1: Yes

2. Has the statistical analysis been performed appropriately and rigorously?

Reviewer #1: Yes

3. Have the authors made all data underlying the findings in their manuscript fully available?

Reviewer #1: Yes

4. Is the manuscript presented in an intelligible fashion and written in standard English?

Reviewer #1: Yes

5. Review Comments to the Author

Please use the space provided to explain your answers to the questions above. You may also include additional comments for the author, including concerns about dual publication, research ethics, or publication ethics.

Reviewer #1: I thought this was a really interesting paper that accomplishes the major goal of creating a publicly accessible 3D database of the Rano Raraku quarry site on Rapa Nui—and through the collection of high-resolution imagery—postulates how the famous moai statues were carved and the sociopolitical mechanisms behind their construction. Overall, I thought the paper was well written and have only a few, mostly minor, comments that I think would help to improve the paper.

Abstract – the authors pose the paradox first without presenting the underlying question behind it. Logically, it would make more sense to flip the first and second sentences and rephrase slightly.

Agreed. We have revised the abstract to improve clarity.

While the authors provide a brief background section on Rapa Nui and the quarry, for readers unfamiliar with anything but what these statues look like, it would be helpful to provide a brief summary on what we know archaeologically about the colonization of Rapa Nui and the major cultural stages. Included in this should be some temporal inferences that provide context on when the island was first settled, when quarrying began (and at what quarries) and ended, where these statues were eventually placed around the island, etc. This doesn’t have to be lengthy (and the paper is relatively short, so there would seem to be plenty of room to add this in), and some kind of cultural chronology would help many readers (re)familiarize themselves with Rapa Nui’s history which the authors seem to take for granted that everyone knows.

Agreed. We have added text to place the island into the context of the archaeological record of Rapa Nui.

In terms of the centralization question and its role in moai manufacture, I had a few comments/queries. The first is, even in a centralized organization, can there not be variation in what is produced? The authors make the argument that what the detailed imagery (and previous inferences from other scholars) shows is not a hierarchical nature but one that is instead heterarchical. That may very well be the case, but there are many past and present analogues that demonstrate variable ways for producing or constructing objects (apart from mass production where there are templates/casts, etc.).

Agreed. We have added a comment about variation in moai production, whether centrally organized or not.

For example, in many societies that have centralized power (whether big or small), the ways in which megaliths, pottery, stone tools, textiles, etc. were produced can have inherent variation in style and decoration that is reflective of family units, clan/kin, or artisanry groups. I guess that I do not necessarily see the impossibility of this in the Rapa Nui case, despite other lines of evidence suggesting that decentralized power was perhaps the norm.

p. 10 – what is meant by “small production teams”? Are we talking 3 people? 10? 25? And while the workshop area might have certainly limited those who had direct access to the statue being carved, I would imagine that others would be in line to take over during rest breaks, etc. Production crews would also presumably involve others who would provide resources (food, water), etc. I guess my point here is, though the physical space to carve might be constrained, this does not necessarily equate to “smaller production teams” which could be X number of times bigger.

Agreed. We have revised the text to include comments about the potential number of people.

p. 11 – back to the question of centralized coordination and “relatively small groups”, what is meant by “small” and cannot centralization be at the level of smaller (micro) or medium (meso) sized groups (however defined)? I guess the issue for me is that the authors are painting a picture of centralization that assumes an island-wide centralized effort, but a strict definition of the term does not require a large island-wide/village sized group as a requirement for centralized power. Hypothetically, this requirement could technically be met by one village of 100 people or less consolidating their power to quarry one part of Rano Raraku and a completely separate village doing the same contemporaneously. Each has centralized their efforts, which the same evidence presented in this paper would seem to show.

To remedy this, I think the authors should provide a more detailed explanation of what they mean by centralization and connect this to an expanded background section on Rapa Nui prehistory which could provide some additional information on where villages were located, how many the authors think were occupied at any one time, and by how many people.

Agreed. We have revised the text and added clarifications.

p. 11 – authors propose “high local cultural retention with limited long-range community interaction.” On an island that is only about 23 km across, what is meant here by “long-range”? This of course is a relative scale, but when it comes down to communication, this would seem to be a negligible variable.

Agreed. We have edited the text for clarity on “long range.” (We meant island-wide).

Finally, are there any corollaries in the Pacific or elsewhere that the authors could provide which would either provide support for their interpretation of societies that can “effectively coordinate large-scale productive activities through distributed networks of semi-autonomous groups, while maintaining both competitive dynamics and cultural cohesion”. It would be helpful for readers to know whether this is a highly unique feature to Rapa Nui, or something that is visible elsewhere (and in what circumstances/contexts).

Agreed. We have added a couple of well-documented corollaries for large-scale productive works that emerged out of the activities of semi-autonomous groups. The examples are from Hawai`i Island agricultural field systems in Kona and Kohala (with works cited).

---

## [Editor Report · Decision Letter 1]

22 Oct 2025

Megalithic statue (moai) production on Rapa Nui (Easter Island, Chile)

PONE-D-25-37143R1

Dear Dr. Lipo,

We’re pleased to inform you that your manuscript has been judged scientifically suitable for publication and will be formally accepted for publication once it meets all outstanding technical requirements.

Kind regards,

Peter F. Biehl, PhD

Academic Editor

PLOS ONE
---

## [Editor Report · Acceptance letter]

PONE-D-25-37143R1

PLOS ONE

Dear Dr. Lipo,

I'm pleased to inform you that your manuscript has been deemed suitable for publication in PLOS ONE. Congratulations! Your manuscript is now being handed over to our production team.

Kind regards,

on behalf of

Dr. Peter F. Biehl

Academic Editor

PLOS ONE